# Ximaoornatins A–C, Polyoxygenated Diterpenoids from the Hainan Soft Coral *Sinularia ornata*

**DOI:** 10.3390/md20030218

**Published:** 2022-03-20

**Authors:** Li-Li Sun, Xu-Wen Li, Yue-Wei Guo

**Affiliations:** 1State Key Laboratory of Drug Research, Shanghai Institute of Materia Medica, Chinese Academy of Sciences, 555 Zu Chong Zhi Road, Zhangjiang Hi-Tech Park, Shanghai 201203, China; sunlili202105@126.com; 2University of Chinese Academy of Sciences, No. 19A Yuquan Road, Beijing 100049, China; 3Drug Discovery Shandong Laboratory, Bohai Rim Advanced Research Institute for Drug Discovery, Yantai 264117, China

**Keywords:** soft coral, *Sinularia ornata*, polyoxygenated diterpenoid, stereochemistry, activity

## Abstract

Three complex polyoxygenated diterpenoids possessing uncommon tetradecahydro-2,13:6,9-diepoxybenzo[10]annulene scaffold, namely ximaoornatins A–C (**1**–**3**), one new eunicellin-type diterpene, litophynin K (**4**), and a related known compound, litophynol B (**5**) were isolated from the South China Sea soft coral *Sinularia ornata*. The structures and absolute configurations of **1**–**4** were established by extensive spectroscopic analysis, X-ray diffraction analysis, and/or modified Mosher’s method. A plausible biosynthetic relationship of **1** and its potential precursor **4** was proposed. In a bioassay, none of the isolated compounds showed obvious anti-inflammatory activity on LPS-induced TNF-α release in RAW264.7 macrophages and PTP1B inhibitory effects.

## 1. Introduction

The soft corals of the genus *Sinularia* (phylum Cnidaria, class Alcyonaria, subclass Octocorallia, order Alcyonacea, family Alcyoniidae), were widely distributed in oceans including the South China Sea. Chemical investigations indicated that *Sinularia* corals are rich sources of diverse and complex metabolites (terpenoids, steroids, polyamines, alkaloids, *etc.*), with widespread biological activities, such as immunological, cytotoxic, antibacterial, and anti-inflammatory properties [1,2,3,4]. A literature survey disclosed that, among all the species of *Sinularia*, *S. ornata* have never been chemically studied.

In our ongoing search aiming to explore bioactive secondary metabolites from Chinese marine organisms [5,6,7,8,9], *S. ornata* were collected off Ximao Island, Hainan Province, China, and chemically investigated, resulting in the discovery of three structurally unprecedented diterpenoids, namely ximaoornatins A–C (**1**–**3**), one new eunicellin-type diterpene, litophynin K (**4**), and a related known compound, litophynol B (**5**) [10] (Figure 1). Detailed isolation, full structural determination, as well as the plausible biosynthetic pathway of the new compounds are reported herein.

## 2. Results and Discussion

The acetone extract of the soft coral *S. ornata* was partitioned between Et_2_O and H_2_O. The Et_2_O-soluble portion was subjected to repeated chromatography over silica gel, Sephadex LH-20, and RP-HPLC to afford four new compounds, compounds **1** (11.6 mg), **2** (1.2 mg), **3** (2.0 mg), **4** (12.3 mg) and **5** (17.0 mg), respectively.

Compound **1** was isolated as an optically active colorless crystal. Mp. 166.2–166.4 °C. Its molecular formula was deduced to be C_26_H_40_O_7_ by the HRESIMS 487.2670 [M + Na]^+^ (cald 487.2666), indicating the presence of seven degrees of unsaturation. The ^13^C NMR, DEPT and HSQC spectra (Appendix A) disclosed 26 carbon signals, including six methyls, five sp^3^ methylenes, nine sp^3^ methines (five oxygenated ones at *δ*_C_ 75.7, 78.4, 78.9, 86.4, and 86.8), two oxygenated sp^3^ quaternary carbons (*δ*_C_ 79.1, 85.2), one sp^2^ methylene, three sp^2^ quaternary carbon (two ester carbonyls at *δ*_C_ 170.2, 173.0). The diagnostic ^1^H and ^13^C NMR resonances (Appendix A), as well as coupling constants of the connected protons, revealed the presence of one disubstituted terminal double bond (*δ*_H_ 5.08, s, 1H, H-17a; *δ*_H_ 5.05, s, 1H, H-17b; *δ*_C_ 111.5, CH_2_, C-17; 144.6, C, C-11). One double bond and two ester carbonyls accounted for three of the total seven degrees of unsaturation, implying a tetracyclic ring system in the molecule.

The structure of **1** was established by detailed 2D NMR analysis (Figure 2). The extensive analysis of the ^1^H−^1^H COSY spectrum (Appendix A) of **1** elucidated four structural fragments **a**–**d**, by clear correlations of H_2_-4 (*δ*_H_ 3.01, 1.45)/H_2_-5 (*δ*_H_ 1.50, 1.44)/H-6 (*δ*_H_ 3.90) (**a**); H-8 (*δ*_H_ 5.27)/H-9 (*δ*_H_ 4.62)/H-10 (*δ*_H_ 2.57)/H-1 (*δ*_H_ 2.65) (**b**); H_2_-12 (*δ*_H_ 2.45, 2.38)/H-13 (*δ*_H_ 4.10)/H-14 (*δ*_H_ 1.91)/H-1 (*δ*_H_ 2.65)/H-2 (*δ*_H_ 4.39), H-14/H-18 (*δ*_H_ 1.79)/H_3_-19 (*δ*_H_ 1.07) and H-18/H_3_-20 (*δ*_H_ 1.00) (**c**); H_2_-2′ (*δ*_H_ 2.26)/H_2_-3′ (*δ*_H_ 1.66)/H_3_-4′ (*δ*_H_ 0.97) (**d**), respectively. Fragments **a** and **b** were deduced to be connected through C-7 by the HMBC correlations (Appendix A) from H_3_-16 to C-6/C7/C-8. The HMBC correlations from H_2_-17 to C-10/C-11/C-12 and from H-10 to C-1/C-2/C-12 determined the presence of a cyclohexane ring (ring A) with a terminal double bond at C-11 and an isopropyl group at C-14. The cross peaks from H_3_-15 to C-2/C-3/C-4, revealed that the fragments **a** and **c** were connected via the quaternary carbon C-3. Furthermore, the presence of ether bridges between C-2 and C-13 and between C-6 and C-9 were deduced by HMBC correlations from H-13 to C-2 and from H-6 to C-9, respectively, forming two tetrahydrofuran rings (rings B and D) and one nonatomic ring C. Finally, the presence of a butyryloxy group at C-3 was deduced by the diagnostic HMBC correlations from H_2_-2′ to C-1′ and from H-2 to C-1′. In addition, the key HMBC correlations from H-8 to C-1″ and from H_3_-2″ to C-1″ indicated the connection of an acetyloxy group at C-8. Thus, the structure of **1** was determined as drawn in Figure 2, with an uncommon tetracyclic ring system.

The relative configuration of **1** was determined by a detailed analysis of its NOESY spectrum (Appendix A). As show in Figure 2, the NOE correlations of H-1/H-10, H-1/H-13, H-8/H-10, and H-8/H_3_-16 suggested that H-1, H-8, H-10, H-13, and H_3_-16 were all co-facial, arbitrarily assigned as *β*-configuration. The opposite (*α*) orientation of H-2, H-6, H-9, H-14, and H_3_-15 was indicated by the NOE cross peaks of H-2/H-9, H-2/H-14, H-2/H_3_-15, and H-6/H-9. Finally, the relative configuration of **1** was established as 1*R**,2*S**,3*R**,6*R**,7*R**,8*S**,9*S**,10*R**,13*R**, 14*R**.

To determine the absolute configuration of **1**, we fortunately managed to obtain its suitable single crystals in methanol, which were successful applied on X-ray crystallography using Cu K*α* radiation (*λ* = 1.54178 Å). The analysis of the X-ray data not only unambiguously confirmed the structure of **1** but also disclose its absolute configuration as 1*R*,2*S*,3*R*,6*R*,7*R*,8*S*,9*S*, 10*R*,13*R*,14*R* (Flack parameter was −0.09(7)) (Figure 3, CCDC 2126980).

Compound **2** was isolated as colorless oil with the chemical formula of C_28_H_44_O_7_ as disclosed by the HREIMS ion peak at *m/z* 492.3080, ([M]^+^, calcd 492.3082), implying seven degrees of unsaturation. The ^13^C NMR, DEPT and HSQC spectra (Appendix A) revealed the presence of 28 carbons including six methyl groups, eight methylenes, nine methines, three quaternary carbons and two ester carbonyls (*δ*_C_ 172.8, 172.9). In fact, compound **2** displayed very similar 1D NMR data as those of **1** (Table 1), with the only difference on the substitution at C-8 position. Instead of the acetoxyl group in **1**, the HMBC correlations (Appendix A) from H-8 to 1″ and the ^1^H-^1^H COSY correlations (Appendix A) of H_2_-2″ (*δ*_H_ 2.38)/H_2_-3″ (*δ*_H_ 1.70)/H_3_-4″ (*δ*_H_ 0.98), indicating the butyryloxy group at the C-8 of **2**, which was in agreement with a 28 mass units’ difference in their molecular weights. Therefore, the structure of **2** was determined as shown in Figure 1, named ximaoornatin B.

Compound **3** was isolated as an optical active colorless oil. From the molecular ion peak at *m/z* 422.2665 [M]^+^ (calcd 422.2663) in the HRESIMS spectrum, a molecular formula of C_24_H_38_O_6_ was elucidated, indicating six degrees of unsaturation. The 1D NMR data of **3** were reminiscent of those of **1** and **2** (Table 1), and a further analysis of their 2D NMR spectra (Figure 2, Appendix A) suggested the same skeleton of the three compounds with the same tetrahydrofuran rings A−D. The main differences between these compounds were found to be the presence of a hydroxy group at C-8 (*δ*_C_ 79.0; *δ*_H_ 4.09) in **3** instead of the acetoxyl group in **1** and butyroxyl group in **2**, which was also confirmed by their mass spectrum. Thus, compound **3** was C-8 deacetyl derivative of **1**, named ximaoornatins C.

The relative configurations of **2** and **3** were assigned to be the same as that of **1** due to the same NOE patterns in all three compounds. The absolute configurations of **2** and **3** were also assigned to be same as that of **1** by comparing their NMR spectra and on a biogenetic consideration since they only differed by the different substitution on 8-OH.

The molecular formula of compound **4** was found to be C_26_H_40_O_6_ by HRESIMS (*m/z* 471.2718 [M + Na]^+^, calcd 471.2717), suggesting seven degrees of unsaturation. The ^13^C NMR and HSQC spectra (Appendix A) disclosed the presence of 26 carbons including six sp^2^ carbon atoms (2 × C=O, 1 × C = CH_2_, 1 × C = CH) at lower field and twenty sp^3^ carbon atoms at higher field (4 × OCH, 1 × OC, 5 × CH_2_, 4 × CH, 6 × CH_3_), accounting for four degrees of unsaturation. Thus, the remaining three degrees of unsaturation reveal **4** as a tricyclic molecule. Detailed analysis of its NMR data indicated that the spectroscopic features of **4** were similar to those of the known compound litophynin B (**6**) [11]. The apparent downfield shift of H-13 (from *δ*_H_ 1.75, 1.05 in **6** to *δ*_H_ 3.57 in **4**) and C-13 (from *δ*_C_ 25.4 in **6** to *δ*_C_ 72.3 in **4**) indicated the presence of 13-OH in **4**, which was implied by the clear COSY correlations (Appendix A) of H_2_-12 (*δ*_H_ 2.46, 2.31)/H-13/H-14 (*δ*_H_ 1.40). In addition, the significant HMBC cross-peaks (Appendix A) from H-8 (*δ*_H_ 4.83) to C-1″ (*δ*_C_ 170.6)/C-2″ (*δ*_C_ 21.4), and from H-2″ to C-1″ confirmed the replacement of a butyryloxy group at C-8 in litophynin B by an acetyloxy group in **4**. Therefore, the structure of **4** was defined as shown in Figure 1.

The relative configuration of **4** was confirmed by NOESY experiment (Figure 2 and Appendix A). The NOE correlations (Figure 2) of H-5/H_3_-16 proved that the Δ^6,7^ double bond has *E*-geometry. The NOE correlations between H-1 (*δ*_H_ 2.03) and H-10 (*δ*_H_ 2.82), H-1 and H-13, and H-8 and H-10, suggested that H-1, H-8, H-10, and H-13 were *β*-oriented. The correlations of H-2 (*δ*_H_ 3.86)/H-14, H-2/H_3_-15 (*δ*_H_ 1.47), and H-9/H-14, suggested that of all of H-2, H-9 (*δ*_H_ 4.09), H-14, and H_3_-15 are *α*-oriented. From the above evidence, the relative configuration of **4** was determined as 1*R**,2*R**,3*R**,8*R**,9*S**,10*R**,13*R**,14*R**. Finally, to deduce the absolute configuration of the secondary alcohol at C-13, two aliquots of compound **4** were treated with (*R*)- and (*S*)-*α*-methoxy-*α*-trifluoromethylphenyl acetyl (MTPA) chlorides to obtain the (*S*)- and (*R*)-esters, respectively. The analysis of Δ*δ^SR^* values (*δ^S^*−*δ^R^*) observed for the signals of the protons close to 13-OH and according to Mosher’s rule [12,13], the absolute configuration at C-13 in **4** was established as *R* (Figure 4). Thus, the stereochemistry of **4** was unambiguously elucidated as 1*R*,2*R*,3*R*,8*R*,9*S*,10*R*,13*R*,14*R*.

Compounds **1**–**3** comprise an unprecedented tetradecahydro-2,13:6,9-diepoxybenzo[10]annulene skeleton, which were totally different from the co-occurring compound **4** and other eunicellin-type diterpenes. However, they are structurally related by sharing some common moieties, such as a six-membered ring A. Therefore, a plausible biosynthetic connection from the eunicellin diterpene **4** to **1** was proposed as drawn in Figure 1, which mainly underwent an oxidation of Δ^6,7^ on **4** towards the epoxide intermediate **4a**, followed by an acid-promoted electron delivery from the 13-hydroxyl to 6,7-epoxyl via the 2,9-ether group. All the isolates were subjected to the test of anti-inflammatory effects on LPS-induced TNF-α release in RAW264.7 macrophages and PTP1B inhibitory effects, and none of them showed obvious activities.

## 3. Materials and Methods

### 3.1. General Experimental Procedures

IR spectra were recorded on a Nicolet 6700 spectrometer (Thermo Scientific, Waltham, MA, USA); peaks are reported in cm^−1^. Melting points were measured on an X-4 digital micro-melting point apparatus. Optical rotations were measured on a Perkin-Elmer 241MC polarimeter (PerkinElmer, Fremont, CA, USA). The NMR spectra were measured at 300 K on DRX 500 and Avance 600 MHz NMR spectrometers (Bruker Biospin AG, Fallanden, Germany). Chemical shifts are reported in parts per million (*δ*) in CDCl_3_ (*δ*_H_ reported referred to CHCl_3_ at 7.26 ppm; *δ*_C_ reported referred to CDCl_3_ at 77.16 ppm) and coupling constants (*J*) in Hz; assignments were supported by ^1^H–^1^H COSY, HSQC, HMBC, and NOESY experiments. HR-ESIMS was carried out on a Waters Q-TOF Ultima mass spectrometer (Waters, MA, USA). HREIMS spectra were carried out on a Thermo DFS mass spectrometer. Semi-preparative HPLC was performed on an Agilent-1260 system equipped with a DAD G1315D detector using ODS-HG-5 (250 mm × 9.4 mm, 5 µm) by eluting with the CH_3_OH–H_2_O or CH_3_CN–H_2_O system at 3.0 mL/min. Commercial silica gel (200−300 and 300−400 mesh; Qingdao, China) was used for column chromatography (CC). Precoated Si gel plates (Merck Chemicals Co., Ltd., G60 F254, Shanghai, China) were used for analytical TLC. Sephadex LH-20 (Amersham Biosciences, London, U.K.) was also used for CC. All solvents used for column chromatography and HPLC were of analytical grade (Shanghai Chemical Reagents Co., Ltd., Shanghai, China) and chromatographic grade (Dikma Technologies Inc., Shanghai, China), respectively. X-ray diffraction studies were carried out on a Bruker D8 Venture diffractometer with Cu Kα radiation (λ = 1.54178 Å).

### 3.2. Biological Materials

Specimens of the soft coral *Sinularia ornata*, identified by Prof. Xiu-Bao Li from Hainan university, were collected along the coast of Ximao Island, Hainan province, China, in 2018, and were frozen immediately after collection. A voucher specimen (18-XD-07) was deposited at the Shanghai Institute of Materia Medica, Chinese Academy of Sciences, Shanghai, China.

### 3.3. Extraction and Isolation

The frozen materials (943 g, dry weight) were cut into pieces and exhaustively extracted with Me_2_CO at room temperature. The organic extract was evaporated to give a brown residue, which was partitioned between Et_2_O and H_2_O. The Et_2_O solution was concentrated under reduced pressure to give a dark brown residue (36.3 g), which was fractionated by gradient Si gel (200−300 mesh) column chromatography (CC) (0 → 100% Et_2_O in petroleum ether (PE), yielding eight fractions (A−F). Fraction D was isolated by Sephadex LH-20 (PE/CH_2_Cl_2_/MeOH, 2:1:1), followed by silica gel CC (PE/CH_2_Cl_2_, 10:0 → 0:10) to give two subfractions (D2E and D2G). Fraction D2E was finally purified by reversed-phase HPLC (MeCN/H_2_O, 70:30; 3.0 mL/min) to give compound **1** (11.6 mg, t_R_ = 17.1 min) and **4** (12.3 mg, t_R_ = 12.3 min), while compound **2** (1.2 mg, t_R_ = 18.6 min) was isolated from subfraction D2G by RP-HPLC (MeCN/H_2_O, 60:40; 3.0 mL/min). Fraction E was fractioned by Sephadex LH-20 (PE/CH_2_Cl_2_/MeOH, 2:1:1), followed by silica gel CC (PE/Et_2_O, 10:0 → 0:10) to give two subfractions (E3E and E3H). Subfraction E3E was further purified by stepwise HPLC (MeCN/H_2_O, 62:38 → 70:30; 3.0 mL/min) to obtain compound **5** (17.0 mg, t_R_ = 5.3 min). Similarly, compound **3** (2.0 mg, t_R_ = 13.5 min) was isolated from fraction E3H by reversed-phase HPLC (MeCN/H_2_O, 75:25 → 98:2; 3.0 mL/min).

Ximaoornatin A (**1**): colorless crystals; m.p. 166.2~166.4 °C; [*α*]${}_{D}^{20}$−46.7 (*c* 0.35, CHCl_3_); IR (KBr) *ν*_max_ = 3445, 2919, 2849, 1959, 1620, 1384, 1156, 1043 cm^−1^; ^1^H and ^13^C NMR data see Table 1; HR-ESIMS *m/z* 487.2670 [M + Na]^+^ (calcd. for C_26_H_40_NaO_7_, 487.2666).

Ximaoornatin B (**2**): colorless oil; [*α*]${}_{D}^{20}$−15.4 (*c* 0.12, CHCl_3_); IR (KBr) *ν*_max_ = 3441, 2960, 2924, 2870, 1959, 1732, 1620, 1384, 1260, 1074, 1040 cm^−1^; ^1^H and ^13^C NMR data see Table 1; HR-EIMS *m/z* 492.3080 [M]^+^ (calcd. for C_28_H_44_O_7_, 492.3082).

Ximaoornatin C (**3**): colorless oil; [*α*]${}_{D}^{20}$−55.0 (*c* 0.11, CH_3_OH); IR (KBr) *ν*_max_ = 3444, 2924, 1959, 1731, 1620, 1384, 1260, 1045, 800 cm^−1^; ^1^H and ^13^C NMR data see Table 1; HR-EIMS *m/z* 422.2665 [M]^+^ (calcd. for C_24_H_38_O_6_, 422.2663).

Litophynin K (**4**): colorless oil; [*α*]${}_{D}^{20}$−58.4 (*c* 0.24, CHCl_3_); IR (KBr) *ν*_max_ = 3446, 2925, 1959, 1733, 1665, 1384, 1247, 1097, 1051 cm^−1^; ^1^H and ^13^C NMR data see Table 1; HR-ESIMS *m/z* 471.2718 [M + Na]^+^ (calcd. for C_26_H_40_NaO_6_, 471.2717).

X-ray Crystallographic Analysis for **1**. Colorless blocks, C_2.08_H_3.2_O_0.56_, *M*_r_ = 37.17, monoclinic, crystal size 0.15 × 0.08 × 0.05 mm^3^, space group *P*2_1_, *a* = 11.0532(5) Å, *b* = 9.4225(4) Å, *c* = 12.3134(5) Å, *V* = 1271.54 (9) Å^3^, *Z* = 25, *D*_calcd_ = 1.213 g/cm^3^, *F*(000) = 504.0, 19,814 reflections measured (7.24° ≤ 2Θ ≤ 149.14°), 5096 unique (*R*_int_ = 0.0470, *R*_sigma_ = 0.0381) which were used in all calculations. The final *R*_1_ was 0.0346 (*I* > 2*σ*(*I*)) and w*R*_2_ was 0.0901 (all data). The X-ray measurements were made on a Bruker D8 Venture X-ray diffractometer with Cu K*α* radiation (*λ* = 1.54178 Å) at 170.0 K. The structure was solved with the ShelXT structure solution program using Intrinsic Phasing and refined with the ShelXL refinement package using least squares minimization. Crystallographic data for **1** were deposited at the Cambridge Crystallographic Data Centre (Deposition nos. CCDC 2126980). Copies of these data can be obtained free of charge via www.ccdc.cam.ac.uk/conts/retrieving.html or from the Cambridge Crystallographic Data Centre, 12 Union Road, Cambridge CB21EZ, UK (fax: +44-1223-336-033; e-mail: deposit@ccdc.cam.ac.uk).

### 3.4. Anti-Inflammatory Activity Assay

RAW264.7 cell, a murine macrophage cell line, was obtained from American Type Culture Collection (ATCC, Manassas, VA, USA). In the bioassay for anti-inflammation, RAW264.7 cells were grown in DMEM containing 2 mmol/L L-glutamine, 10% FBS, 100 U/mL penicillin, and 100 μg/mL streptomycin, and maintained in a humidified incubator of 5% CO_2_ at 37 °C. The anti-inflammatory effect was measured by the cell viability and TNF-α production of RAW264.7 cells. The cells (1 × 10^5^/well) were incubated in 96-well plates in triplicate. For the cell viability part, RAW264.7 cells were cultured with vehicle (final concentration of 0.125% DMSO) or tested compounds at the indicated concentrations for 24 h. A total of 20 μL CCK-8 reagent was added to each well and after 1 h incubation and the OD values were collected after 1 h incubation at 450 nm (650 nm calibration) by a microplate reader (Molecular Devices, Sunnyvale, CA, USA). For the anti-inflammatory activity assay, after adherence, the cells were cultured with vehicle (final concentration of 0.125% DMSO) or tested compounds at the indicated concentrations for 30 min. Then, the cells were primed with 1 μg/mL of LPS (Lipopolysaccharide) for 24 h. Supernatants were centrifuged and then quantified with the mouse TNF-α ELISA kit following the manufacturer’s instructions. The CC50 and IC50 were estimated using the log (inhibitor) vs. normalized response nonlinear fit (Graph Pad Prism 6.0).

### 3.5. PTP1B Inhibitory Activity Assay

The recombinant PTP1B catalytic domain was expressed and purified according to a previous report [13]. The enzymatic activities of the PTP1B catalytic domain were determined at 30 °C by monitoring the hydrolysis of *p*NPP. The dephosphorylation of *p*NPP generates product *p*NP, which was monitored at an absorbance of 405 nm by the EnVision multilabel plate reader (PerkinElmer Life Sciences, Boston, MA, USA). In a typical 100 μL assay mixture containing 50 mmol/L 3-[N-morpholino]-propanesulfonic acid (MOPs), pH 6.5, 2 mmol/L *p*NPP, and 30 nmol/L recombinant PTP1B, activities were continuously monitored and the initial rate of the hydrolysis was determined using the early linear region of the enzymatic reaction kinetic curve. The IC50 was calculated with Prism 4 software (Graphpad, San Diego, CA, USA) from the nonlinear curve fitting of the percentage of inhibition (% inhibition) vs. the inhibitor concentration [I] using the following equation: % inhibition = 100/(1 + [IC50/[I]]*^k^*), where *k* is the Hill coefficient.

## 4. Conclusions

In summary, this is the first detailed chemical investigation of the soft coral *S. ornata* leading to the isolation and full characterization of three unusual diterpenoids, ximaoornatins A–C (**1**–**3**), with unprecedented tetradecahydro-2,13:6,9-diepoxybenzo[10]annulene skeleton, and a related new eunicellin-type diterpene, litophynin K (**4**). The two ether bridges between C-2 and C-13, and between C-6 and C-9, respectively, as well as the tetracyclic ring systems of compounds **1**–**3**, are totally different from the skeleton of eunicellin diterpenoids, the 2,11-cyclized cembranoids [14]. Therefore, compounds **1**–**3** represent a new subclass of 2,11-cyclized cembranoids which have greatly expanded the diversity and complexity of the marine diterpenoids. The stereochemistry of the new compounds was determined by extensive spectroscopic analysis, X-ray diffraction analysis, and/or modified Mosher’s method. In the bioassay, none of the isolates showed obvious anti-inflammatory activity and PTP1B inhibitory effects. Further biomimetic synthesis and other biological studies should be conducted to understand the real ecological and/or biological role played by these interesting molecules during the life cycle of the soft corals and their possible medicinal application.

## Data Availability

Data are contained within the article or Appendix A.

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
