# Peer review of "Ximaoornatins A–C, Polyoxygenated Diterpenoids from the Hainan Soft Coral Sinularia ornata"

_marinedrugs, 2022, doi:10.3390/md20030218_

Round 1

Reviewer 1 Report

See attached file

Author Response

This article describes the isolation and structural elucidation of four new diterpenic substances from the soft coral Sinularia ornate. The spectroscopic characterization and interpretation, including an X-Ray analysis of one of the supports the proposed structures. A reasonable, possible, biosynthetic correlation between them is suggested on the base of the chemical structures. Therefore, I consider that this paper should be published in the Journal Marine Drugs with minor corrections.

Response: We are very grateful for the reviewer’s positive comments and nice suggestions.

Page 4, figure 2.

The structure of compound 6 does not match the structure of litophynin B as described in reference 11 (Ochi et al, Chem. Lett. 1987). In that reference, litophynin B has the molecular formula C28H44O5, due to the presence of another butyroxyl residue on C8. In figure 2, an acetoxy group is drawn on C8 of compound 6. The figure should be properly modified.

Response: Thanks for the referee’s constructive suggestion. We are sorry for the carelessness, and the mistake was corrected accordingly.

Page 4, figure 2.

I think that the 3D bottom drawings in figure 2 need higher resolution in the skeletal backbone of compounds 1, 2, 3 and 4. Although the arrows are perfectly clear and have good resolution, the molecule backbones do not have enough resolution. In order to be seen properly, a normal person requires a large zoom magnification of the file, and the resolution is not enough to give a clear image of the 3-D shapes of these molecules.

Response: We highly appreciate the referee’s correction, and the pointed details were revised accordingly and highlighted in yellow in the manuscript.

Page 4, figure 3

Again this figure resolution is too low. A higher resolution is required to appreciate the stereochemistry of the molecule.

Response: Thanks for your kind suggestion. We have provided a higher resolution figure.

Page 5, line 106.

Replace “indicating seven degrees of unsaturation” by “indicating six degrees of unsaturation”

Response: We highly appreciate your constructive suggestion, and we are sorry for the careless mistake, which have been revised.

Page 5, line 136. This is a language style suggestion:

Replace “proved that Δ6,7 was both of E-geometry” by “proved that Δ6,7 double bond has E-geometry”

Page 6, sentence in lines 169-171. This is a language style suggestion:

Replace “inhibitory effects, while showed no obvious activities” by “inhibitory effects, although no one showed any obvious activities”

Response: Many thanks for these useful suggestions. We have revised the sentences following your advice.

In addition, I miss in the Materials and Methods (section 4) or in the supplementary information, a description of the procedure for the biological activity tests described in page 6, lines 169-171, as well as a literature reference for that procedure.

Response: Many thanks for the referee’s professional suggestion and we completely agree with your opinions. We have added the details for the biological activity tests in the methods and the results sections.

Reference 4 should include the article number: Mar. Drugs 2021, 19(6), 335. https://doi.org/10.3390/md19060335.

The supporting information file has an index referring to page numbers. However, the document lacks page numbers. Please add page numbers.

Please include a high-resolution high-size structure of compound 1 in section 2.1 (X-ray crystallographic analyses of 1). A proper copy of the figure 3 in the main manuscript, but with better resolution.

Response: Thanks for the referee’s constructive suggestion. They have been revised in the manuscript accordingly.

Reviewer 2 Report

Dear Authors,

The manuscript described the isolation of four new metabolites from Sinularia ornata collected at Ximao Island, where the structures were determined by NMR, x-ray and Mosher's method. No biological activity was reported from these isolated compounds.

1) Figure 1 showed six chemical structures but the maintext mentioned five isolated compounds.

2) Compound 4 in acetone for few days can be converted to compound 1 and 5?

Author Response

Dear Authors,

The manuscript described the isolation of four new metabolites from Sinularia ornata collected at Ximao Island, where the structures were determined by NMR, x-ray and Mosher's method. No biological activity was reported from these isolated compounds.

1) Figure 1 showed six chemical structures but the maintext mentioned five isolated compounds.

2) Compound 4 in acetone for few days can be converted to compound 1 and 5?

Response: We highly appreciate the reviewer’s positive comments and constructive suggestions on our MS.

1) Figure 1 showed six chemical structures but the maintext mentioned five isolated compounds.

Response: Many thanks for the kind suggestion. We isolated compounds 15, and compound 6 was only used as a reference compound for compound 4 in this manuscript.

2) Compound 4 in acetone for few days can be converted to compound 1 and 5?

Response: Many thanks for the advice. Firstly, I don’t think compound 4 could be converted into 5 in acetone, since neither hydrolysis nor dehydroxylation could be easily conducted in such smooth condition. Secondly, as we proposed in the manuscript, the conversion of 4 to 1 should first undergo an oxidation followed by an acid promoted electron delivery, thus it is also not possible to convert 4 into 1 only in acetone.

Reviewer 3 Report

This is a very good paper describing the isolation and identification of four new compounds. Not only is the identification very sound, but it also well described, as are the experimental methods. Supplementary material is also of good quality.

Although there are many comments to the paper, and revisions are required, I think that once improved, it is suitable for publication.

Reviewer 4 Report

The paper by Sun et al. on isolation and structural characterization of new diterpenoids is novel, interesting, and of high quality. There’re a few minor issues that I would like to see addressed before publication. First, in contrast to compounds 1-4, there’s no description of compound 5 in the text, how it was characterized and how its structure was elucidated. This information should be added. Second, there’s a brief statement in the discussion section that biological experiments didn’t show any activity of the new compounds. If these negative results are mentioned, technical details of this experiment should be described in the methods section and the result should be fist presented in the results section, not in the discussion. Finally, there’re no experimental details provided for growth of single crystals for the X-ray study, besides the solvent (methanol). Please add this to the methods section.

Round 2

Reviewer 2 Report

The comments are somewhat been addressed. No further suggestion from me.

Author Response

Review 2

The comments are somewhat been addressed. No further suggestion from me.

Response: We highly appreciate your positive and kind comments

Reviewer 3 Report

General remarks

I believe this is a very good paper and it has improved with the corrections made by the authors.

I have but two minor corrections and I strongly support publication.

Line 30 - Please replace ‘title materials’ with something like ‘samples of S. ornata

Line 55 -  Please remove planar.

Author Response

Review 3

I believe this is a very good paper and it has improved with the corrections made by the authors.

Response: We are very grateful for the reviewer’s positive comments and nice suggestions.

I have but two minor corrections and I strongly support publication.

Line 30 - Please replace ‘title materials’ with something like ‘samples of S. ornata

Line 55 - Please remove planar.

Response: Many thanks for these useful suggestions. We have revised the sentences following your advice.